# Choosing important health outcomes for comparative effectiveness research: 6th annual update to a systematic review of core outcome sets for research

Elizabeth Gargon[1], Sarah L. Gorst[1]*, Karen Matvienko-Sikar[2], Paula R. Williamson[1]

1 Department of Health Data Science, University of Liverpool, Liverpool, United Kingdom, 2 School of Public Health, University College Cork, Cork, Ireland

* sgorst@liverpool.ac.uk

**Data Availability Statement:** All relevant data are within the manuscript and its Supporting information files.

## Abstract

### Background

An annual update to a systematic review of core outcome sets (COS) for research ensures that the COMET database is up-to-date. The aims of this study were to: (i) identify COS that were published or indexed in 2019 and to describe the methodological approaches used in these studies; (ii) investigate whether children have been included as participants in published COS development studies, and which methods have been used to facilitate their participation; iii) update a previous exercise to identify COS relevant to the most burdensome global diseases and injuries.

### Methods

MEDLINE and SCOPUS were searched to identify studies published or indexed between (and inclusive of) January 2019 and December 2019. Automated screening methods were used to rank the citations in order of relevance; the top 25% in ranked priority order were screened for eligibility. COS were assessed against each of the Core Outcome Set-STAndards for Development (COS-STAD). A search of the COMET database was undertaken to identify COS relevant to the 25 leading causes of disease burden.

### Results

Thirty-three studies, describing the development of 37 COS, were included in this update. These studies have been added to the COMET database, which now contains 370 published (1981–2019) COS studies for clinical research. Six (18%) of the 33 studies in this update were deemed to have met all of the minimum standards for COS development (range = 4 to 12 criteria, median = 9 criteria). Of the 370 COS studies published to date, 82 COS have been developed for paediatric health conditions and children would have been eligible to participate in 68/82 of these studies. Eleven of these 68 (16%) COS studies have included children as participants within the development process, most commonly through

**Funding:** Professor Williamson is a National Institute for Health Research (NIHR) Senior Investigator (award number NF-SI_0513-10025). Karen Matvienko-Sikar is supported by a Health Research Board Applying Research into Policy and Practice Fellowship (award number HRB-ARPP-A-011). The funders had no role in study design, data collection and analysis, decision to publish, or preparation of the manuscript.

**Competing interests:** EG and PRW are members of the COMET Management Group. SLG and KMS have declared that no competing interests exist. This does not alter our adherence to PLOS ONE policies on sharing data and materials.

participation in Delphi surveys. Relevant COS were identified for 22/25 leading causes of global disease burden.

## Conclusion

There has been a demonstrated increase in COS developed for both research and routine practice, and consistently high inclusion of patient participants. COS developed for paediatric conditions need to further incorporate the perspectives of children, alongside parents and other adults, and adopt research methods fit for this purpose. COS developers should consider the gaps identified in this update as priorities for COS development.

## Introduction

A core outcome set (COS) for trials is the minimum set of outcomes that should be measured and reported in all clinical trials in a specific condition [1]. The COMET Initiative has identified studies that have developed COS and collated these studies in an online database (http://www.comet-initiative.org/studies/search). An annual update to a systematic review of studies reporting the development of COS for research substantiates the database, maintaining its relevancy and comprehensiveness. The COMET database also includes details of ongoing work relating to COS development. COS development is still a relatively new area of research, and therefore it is important to describe current knowledge and trends about COS development methods to help inform discussion around what COS developers are currently doing; in both what they are doing consistently, what is becoming standard methods of development, what COS developers are doing well, improvements in methods of COS development, and where there might be gaps in health areas or where methodological research is needed. The publication of the annual update to this review of COS is therefore crucial to informing discussion around good methodological practice in the area of COS development.

In previous updates to this review we have highlighted various areas in need of further consideration, particularly the importance of including patient and public participants in the development of COS [2–6]. There has been a trend towards a greater number of studies including patient and public participants, increasing to 71% of COS in update five [6] from 56% in update four [5]. However, research on methods to engage more diverse populations, including areas where patients would benefit from enhanced guidance and support, is needed to further facilitate patient participation in COS development [7]. One such group is children, whose perspectives have been overlooked in COS development to date. Prior to this update, only 13% (8/63) of COS studies developed for paediatric conditions (excluding COS focussed on infants) had input from children [8]. Additionally, COS developers have reported difficulties in engaging and retaining children compared with other stakeholder groups [9–11]. If children are not involved in deciding what outcomes should be included in COS, outcomes important to children are likely to be overlooked.

Minimum standards for COS development methods were established in 2017 [12]. In the previous update to this review, we assessed each of the studies reporting the development of the COS against the COS-STAD criteria and reported that one fifth of the included studies met all of the minimum standards for COS development [6]. We highlighted a particular issue; whereby inadequate descriptions of methods prevented the reader from making an assessment of the a priori status of consensus criteria. Though the time between guideline publication (2017) and the publication of COS included in this current update (2019) is still relatively short, the previous assessment provides a baseline against which a comparison can be made.

The aim of the current study was to update the systematic review of COS to identify COS development studies that were published or indexed in 2019 and to describe the methodological approaches used in these studies. A secondary aim was to explore whether children had been included as participants in COS development studies and to describe the methods used to facilitate their participation. A third aim was to update a previous exercise to identify COS (published or ongoing) that might be relevant to the leading causes of global diseases burden [13].

## Methods

### Systematic review update

The systematic review methods used in this update have been described extensively in previous reports [2–6, 14]. An outline is provided here with expansion on new methods being used for this update.

**Study selection.** The inclusion and exclusion criteria remain unchanged; they were described in full in the original systematic review [14], and updated in the fourth update [5]. Studies were eligible for inclusion if they had applied methodology for determining which outcomes or outcome domains should be measured in clinical trials or health research. Such studies were eligible for inclusion, irrespective of any restrictions by age, health condition or setting [14]. Studies describing the development of a Patient Reported Outcome (PRO) COS (a core set of patient reported symptoms and health related quality of life domains) or Core Event Set (a core set of adverse events or complications) were eligible for inclusion in the review update [5]. Studies describing the update of an existing COS (including studies that have applied methodology for determining how to measure outcomes included in a COS) were included as linked papers to the original COS. Studies that contribute to the development of a COS (e.g. systematic reviews of outcomes, studies eliciting stakeholder group (e.g. patient or clinician) opinion) were included as linked papers to the COS.

The following studies were ineligible for inclusion in the review: studies relating to how, rather than which, outcomes should be measured, except when linked to a COS; studies reporting an overview only with no outcome recommendations; studies relating to outcome recommendations for the assessment of quality or efficiency of care; and studies describing the development of a COS exclusively for clinical practice.

**Identification of relevant studies.** SCOPUS and MEDLINE via Ovid were searched (March 2020) to identify studies that had been published or indexed between, and inclusive of, January 2019 and December 2019. No language restrictions were applied. The search strategy, developed for the original review [14], was used for the current update (S1 Table). Hand searching included studies that had been submitted to the COMET database/website, reference lists in eligible studies, as well as those in ineligible studies that referred to a COS.

**Selecting studies for inclusion in the review.** As previously described [14], records from each database were combined and duplicates removed. Automated screening methods were used to rank the citations, in order of relevance, identified in this update [6, 15, 16]. The cut-off for screening was set to the top 25% of abstracts in ranked priority order [15]. Titles and abstracts of the top 25% ranked citations were screened to assess eligibility (stage 1). The ranked list was ordered alphabetically by author surname, prior to any screening, to avoid rank order bias [6]. The full text of potentially relevant articles were then assessed for inclusion (stage 2). Citations without an abstract could not be ranked and therefore were all screened for eligibility.

Three reviewers (EG, SLG, KMS) independently screened the title and abstract of a third of citations each. All reviewers are experienced in this review and have been involved in previous updates. Each citation was categorised as include, unsure, or exclude. Citations were assessed

by a second reviewer (EG or SLG) when there was any uncertainty; citations were discussed and categorised accordingly. Full papers were retrieved for all abstracts categorised as include or unsure at this stage.

Two reviewers (EG and SLG) independently assessed half of the full papers each for inclusion in the review. As at abstract stage, indecisions at full paper assessment were discussed as necessary, and in cases of disagreement were referred to a third reviewer (PRW). The reasons for exclusion at this stage were documented for articles judged to be ineligible.

**Assessment of COS-STAD minimum standards.** One reviewer (EG or SLG) independently assessed each of the COS development studies against the COS-STAD development standards [12]. A total of 12 criteria representing the 11 minimum standards were assessed in this study. Each criterion was assessed as yes (meeting that standard), no (not meeting that standard) or unsure (it was unclear whether the criteria had been met).

**Data extraction.** As described in full previously [14], data was extracted by one reviewer (EG or SLG) in relation to the study aim(s), health area, target population, interventions covered, methods of COS development and stakeholder groups involved. As described in the previous update, text was extracted to support the COS-STAD assessment being made and to aid discussion where necessary [6].

**Data analysis and presentation of results.** The review is reported in accordance with PRISMA guidelines [17] (S1 Checklist). Studies were described narratively, in text and tables. As previously described, the median and range were presented to summarise the number of the minimum standards met across all of the included COS studies; and percentage frequencies were used to report the number of COS that met each standard [6].

## COS relevant to paediatric health conditions

To address the second aim of this study, we extracted information about whether children had been included as participants in all eligible COS development studies published to date, and noted which methods had been used to facilitate their participation.

## Priority setting

To address the third aim of this study, the 25 most burdensome global diseases and injuries were identified from the updated Global Burden of Disease study [13]. The COMET database was searched (November 2020) for relevant or associated COS (published or ongoing) and were mapped to the leading causes of global disease burden. Data regarding health condition, publication year, scope, participating stakeholder groups and countries involved in the development process were extracted for each identified COS.

## Results

### Description of studies

Through database searching, automated ranking and manual screening, we identified fifty-six records meeting the inclusion criteria (Fig 1). We identified a further 25 records through database alerts and hand searching references. Of these 25 records, 19 were linked supplementary papers providing methodological detail relating to COS studies identified from the database search. Six new studies, which were included following hand searching, had either not been indexed in the bibliographic databases within the appropriate search timeframe or had been incorrectly excluded from an earlier review update.

In total, 81 reports were included in this sixth update of the annual systematic review of COS for research. These reports pertain to 33 new studies, with 29 linked papers, along with

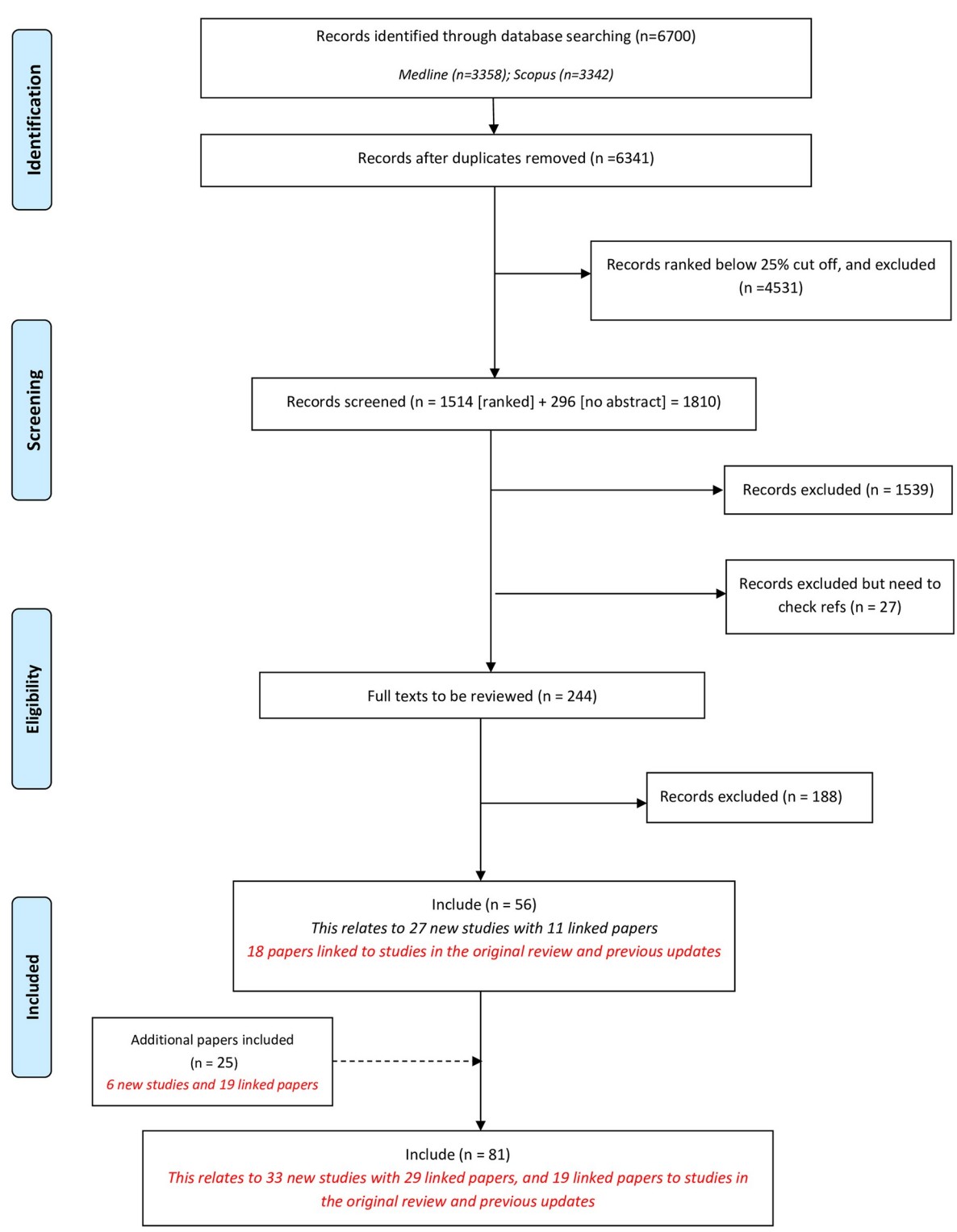

**Fig 1. Identification of studies.**

19 papers linked to studies in the original review and previous updates. The 33 new studies describe the development of 37 individual COS (S2 Table). The sixth update increases the overall total of published COS studies for research to 370, relating to 447 COS.

## Minimum standards

An overview of the minimum standard assessments is provided in Table 1, and by study in S3 Table. Of the 33 COS studies included in this update, six (18%) were deemed to have met all 12 criteria representing the 11 minimum standards for COS development (range = 4 to 12 criteria, median = 9 criteria). All 33 COS studies met all four minimum standards for scope, specifying the setting, health condition, population and intervention covered by the COS. Twenty-three (70%) COS met all three standards for stakeholders involved, including those who will use the COS in research, healthcare professionals and patients or their representatives. Six studies (18%) met all four standards [five criteria] for the consensus process. Thirteen studies (39%) considered both HCPs **and** patients' views when developing this initial list of outcomes. The scoring process and consensus definition were clearly described a priori in fifteen studies (46%) and sixteen studies (49%), respectively. Sixteen studies (49%) described the criteria for including/dropping/adding outcomes a priori. Seventeen studies (52%) took care to avoid ambiguity of language used in the list of outcomes.

## Included studies

Of the 33 new studies identified in this update, 30 were published in 2019 and the remaining three studies were published in 2016, 2017 and 2018 (Fig 2). The COS were developed across 18 disease categories, with 'Orthopaedics and trauma' being the most prevalent area (Fig 3). Details regarding the scope, methods used to develop COS, stakeholders included, and their geographical locations are provided in S4–S7 Tables for the 33 new COS studies included in this update, alongside the six previous systematic reviews. Public participation detail is provided in S8 Table. Details of note include the increase in the number of COS intended for use

**Table 1. COS minimum standards assessments summary (n = 33).**

| DOMAIN | STANDARD NUMBER | STANDARD | STANDARD MET | STANDARD UNCLEAR | STANDARD NOT MET |
|---|---|---|---|---|---|
| | | | N (%) | N (%) | N (%) |
| Scope specification | 1 | The research or practice setting(s) in which the COS is to be applied | 33 (100) | 0 | 0 |
| | 2 | The health condition(s) covered by the COS | 33 (100) | 0 | 0 |
| | 3 | The population(s) covered by the COS | 33 (100) | 0 | 0 |
| | 4 | The intervention(s) covered by the COS | 33 (100) | 0 | 0 |
| Stakeholders involved | 5 | Those who will use the COS in research | 28 (85) | 2 (6) | 3 (9) |
| | 6 | Healthcare professionals with experience of patients with the condition | 32 (97) | 0 | 1 (3) |
| | 7 | Patients with the condition or their representatives | 25 (76) | 0 | 8 (24) |
| Consensus process | 8 | Initial list of outcomes considered both healthcare professionals' and patients' views | 13 (39) | 3 (9) | 17 (52) |
| | 9a | A scoring process was described a priori | 15 (46) | 17 (52) | 1 (3) |
| | 9b | A consensus definition was described a priori | 16 (49) | 16 (49) | 1 (3) |
| | 10 | Criteria for including/dropping/adding outcomes were described a priori | 16 (49) | 16 (49) | 1 (3) |
| | 11 | Care was taken to avoid ambiguity of language used in the list of outcomes | 17 (52) | 15 (46) | 1 (3) |

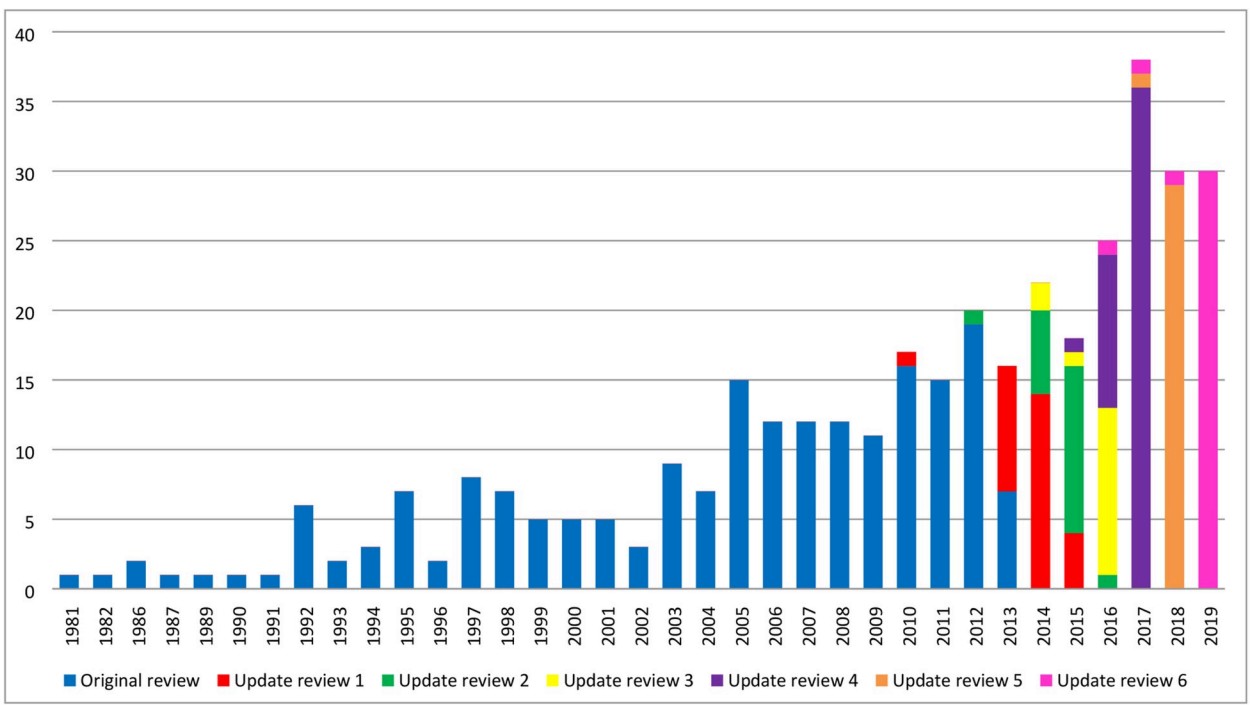

**Fig 2. Year of first publication of each COS study (n = 370).**

in both clinical research and routine clinical practice, which has increased from 11% (36/337) across the studies included in the previous reviews to 27% (n = 9/33) of the studies included in this update (See S4 Table). S7 Table displays the geographical locations of participants included in all published COS development studies. In comparison to the COS studies included in previous reviews, there have been increases in participation from stakeholders located in Asia (65% vs 24%), South America (39% vs 14%) and Africa (32% vs 9%). In addition, fifteen of the 33 new COS studies (45%) included in the current review reported the inclusion of participants from low and middle-income countries (LMICs), as defined by the Organization for Economic Co-operation and Development (OECD) Development Assistance Committee (DAC) list [18].

## COS relevant to paediatric health conditions

Of the 370 COS studies published up until the end of 2019, 82 (22%) are relevant to paediatric health conditions. All studies were published between 1992 and 2019, with 41/82 (50%) studies being published in the past 5 years. The 82 COS studies were developed across 22 disease categories, with neurology (n = 13) and gastroenterology (n = 10) being the most prominent areas. Of the 82 COS studies, 33 (40%) COS are for children, 30 (37%) COS are for adults and children, 14 (17%) COS are for infants and/or toddlers (aged ≤ 4 years), and 5 (6%) COS are for adolescents and adults. Excluding the 14 COS focussed on infants and/or toddlers, there are 68 COS where children (aged ≥ 5 years) could have participated in the development process. Of these 68 COS studies, 11 (16%) have had direct input from children (see Table 2). The methods used in these studies included the Delphi technique, surveys, interviews/focus groups, meetings, and online discussion.

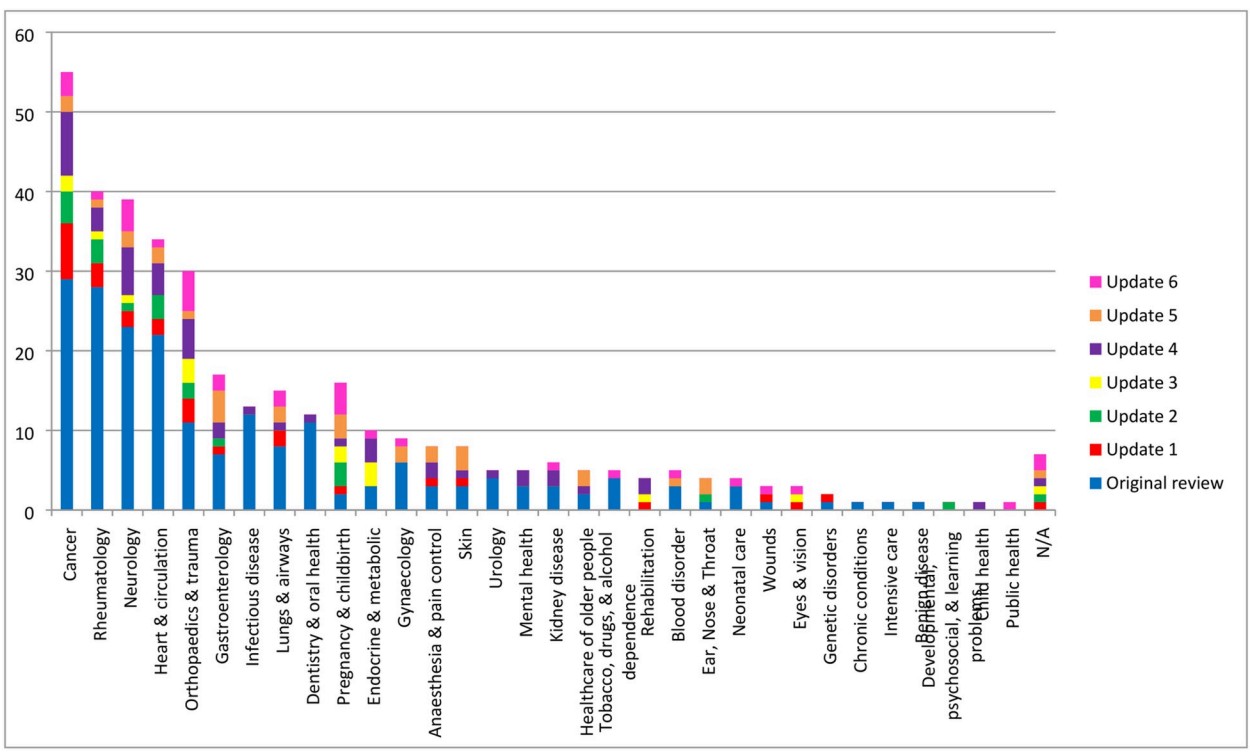

**Fig 3. Number of COS developed in each disease category (n = 370).**

## Priority setting

Twenty-two of the 25 leading causes of global disease burden mapped to at least one relevant published or ongoing COS in the COMET database (see S9 Table): Neonatal disorders, Ischaemic heart disease, Stroke, Lower respiratory infections, Diarrhoeal diseases, Chronic obstructive pulmonary disease (COPD), Diabetes, Low back pain, Congenital birth defects, HIV/ AIDS, Tuberculosis, Depressive disorders, Malaria, Headache disorders, Lung cancer, Chronic

**Table 2. Details of COS including children within the development process.**

| | Year | Disease Category | Disease Name | Population | Methods used |
|---|---|---|---|---|---|
| 1 | 2009 | Neurology | Cerebral palsy | Children | Delphi survey |
| 2 | 2012 | Lungs & airways | Asthma | Children | Survey |
| 3 | 2015 | Developmental, psychosocial, & learning problems | Neurodisability | Children | Interviews; Focus groups; Prioritisation meeting |
| 4 | 2015 | Ear, nose & throat | Cleft palate/ Otitis media | Children | Interviews; Survey |
| 5 | 2016 | Other—healthcare transition | Chronic physical and medical conditions | Children and young adults | Delphi survey; Interviews |
| 6 | 2017 | Gastroenterology | Functional constipation | Children | Delphi survey |
| 7 | 2017 | Gastroenterology | Hirschsprung's disease | Adults and children | Delphi survey |
| 8 | 2017 | Orthopaedics & trauma; Rheumatology | Acute osteomyelitis/ Septic arthritis | Children | Interviews |
| 9 | 2018 | Neurology | Epilepsy | Children | Delphi survey |
| 10 | 2019 | Neurology | Rolandic epilepsy | Children | Delphi survey; Consensus meeting |
| 11 | 1997—updated in 2019 | Rheumatology | Arthritis | Children | Online discussion board |

kidney disease, Other musculoskeletal disorders, Age-related hearing loss, Falls, Gynaecological diseases, Anxiety disorders and Dietary iron deficiency. There were no published or ongoing COS for the remaining three diseases and injuries: Road injuries, Cirrhosis and Self-harm.

## Discussion

Thirty three new studies (relating to 37 COS), were identified and included in this sixth update to the systematic review of COS for research, as well as the COMET database (http://www.comet-initiative.org/studies/search). The annual publication of COS for research is consistent with previous updates [6]. Their inclusion in the COMET database ensures that the database content is both extensive and current.

Six (18%) of the 33 studies in this update met all of the minimum standards for COS development, comparable to the previous update where the minimum standards were considered for the first time and 20% of included studies met all of the minimum standards [6]. The reporting guideline for COS studies [19] was referenced in 39% of studies, less than the 60% in the previous update [6]. One explanation could be that COS developers may not be reporting all of the information necessary to assess whether the minimum standards have been met; alternatively, the COS published in 2019 may have been in progress before the standards were published.

COS can be developed for research, routine care, or both. While the focus of this review is COS for research, there is current interest in identifying whether COS might have a role throughout the healthcare research ecosystem [20]. In the previous update to this review, we highlighted the percentage of COS for research that also intend their recommendations for use in routine care had remained constant at around 11% [6]. The number of COS intended for both research and routine care in this update has increased to 27% of studies. Further research is needed to understand and optimise the methods used to develop COS for multiple settings.

Participation from continents other than Europe and North America continues to increase, including Asia, South America and Africa. Almost half of the studies included participants from LMICs, but this remains an important area for improvement.

The inclusion of public representatives remains high in this review with 76% of studies including this group of stakeholders in the development process. An almost 10% increase in the number of studies that met the minimum standard for taking care around language suggests more developers have considered how to facilitate patient participation. Of particular interest in this update was the participation of children in COS development; of the 68 published COS studies to date that are relevant to paediatric health conditions, 11 have included input from children. Although participation of children is gradually increasing, with nine of these 11 COS being published in the past five years, there is still room for improvement. COS developed for paediatric conditions need to incorporate the perspectives of children, alongside parents and other adults, to ensure that the outcomes measured in both research and clinical practice are ones that matter to children. Key to producing COS that are relevant to children is making sure the research methods used are suitable for them. Across the 11 COS studies that have included children, the methods used have largely been designed for adult participants and it is uncertain whether these are suitable for children. Over recent years, there has been a general increase in COS relating to paediatric conditions, with over 70 currently under development. Thus, there is a pressing need to identify suitable methods for engaging children as participants in COS development studies. Such methods will enable them to meaningfully contribute to the development of COS and ensure that the outcomes measured are ones that matter.

Three of the leading causes of global disease burden, identified from the updated Global Burden of Disease study [13], did not have a potentially relevant COS in the COMET database.

This is an improvement from the first time this exercise was carried out to identify gaps in COS development (carried out in January 2016) where it was reported that 12 COS did not have an applicable COS [3]. In addition to the Global Burden of Disease study, gaps in COS development have been identified in relation to Cochrane systematic reviews, where no published or ongoing COS were found for two of 52 Cochrane Reviews Groups (CRGs) [21]. The current gaps in existing COS should be considered by COS developers to identify priority areas for COS development.

## Conclusion

This annual publication of COS for research has demonstrated an increase in COS developed for both research and routine clinical practice, along with growth in the participation of stakeholders globally. There is a pressing need to identify suitable methods for engaging children as participants in COS development studies. This study can be used by COS developers to help identify gaps and priority areas for COS development.

## Supporting information

**S1 Checklist. PRISMA checklist for content of a systematic review.**
(DOC)

**S1 Table. Search strategy.**
(DOCX)

**S2 Table. Table of reports included in the updated review.**
(DOCX)

**S3 Table. COS minimum standards: Assessment by study (n = 33).**
(DOCX)

**S4 Table. The scope of included studies (n = 370).**
(DOCX)

**S5 Table. The methods used to develop COS (n = 370).**
(DOCX)

**S6 Table. Participant groups involved in selecting outcomes for inclusion in COS (n = 370).**
(DOCX)

**S7 Table. Geographical locations of participants included in the development of each COS (n = 312).**
(DOCX)

**S8 Table. Nature of patient participation where detail is reported (n = 25).**
(DOCX)

**S9 Table. Details about COS relevant to 25 disease with the highest global prevalence.**
(DOCX)

## Acknowledgments

We acknowledge Jane Blazeby (Bristol University) and Mike Clarke (Queen's University Belfast) for their involvement in the conceptualisation and methodology in the original systematic review on which this update is based.

We acknowledge Aurélie Névéol (University Paris-Saclay) and Christopher Norman (University Paris-Saclay, University of Amsterdam) for running the automated screening model and generating the ranked list of citations used in this update.

## Author Contributions

**Conceptualization:** Elizabeth Gargon, Paula R. Williamson.

**Data curation:** Elizabeth Gargon, Sarah L. Gorst, Karen Matvienko-Sikar.

**Formal analysis:** Elizabeth Gargon, Sarah L. Gorst.

**Funding acquisition:** Elizabeth Gargon, Paula R. Williamson.

**Investigation:** Elizabeth Gargon, Sarah L. Gorst.

**Methodology:** Elizabeth Gargon, Paula R. Williamson.

**Writing – original draft:** Elizabeth Gargon, Sarah L. Gorst.

**Writing – review & editing:** Elizabeth Gargon, Sarah L. Gorst, Karen Matvienko-Sikar, Paula R. Williamson.

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
