## [Decision Letter · Decision Letter 0]

4 Sep 2020

PONE-D-20-21185

Choosing important health outcomes for comparative effectiveness research: 6th annual update to a systematic review of core outcome sets for research

PLOS ONE

Dear Dr. Gorst,

Thank you for submitting your manuscript to PLOS ONE. After careful consideration, we feel that it has merit but does not fully meet PLOS ONE’s publication criteria as it currently stands. Therefore, we invite you to submit a revised version of the manuscript that addresses the points raised during the review process.

Please respond to all comments made by the reviewers.

We look forward to receiving your revised manuscript.

Kind regards,

Charles S. Wiysonge, MD, PhD

Academic Editor

PLOS ONE

Journal Requirements:

"Professor Williamson is a National Institute for Health Research (NIHR) Senior Investigator (award number NF-SI_0513-10025). The views expressed in this article are those of the author(s) and not necessarily those of the NIHR, or the Department of Health and Social care. Karen Matvienko-Sikar is supported by a Health Research Board Applying Research into Policy and Practice Fellowship (award number HRB-ARPP-A-011)."

"This work was funded through a National Institute for Health Research (NIHR) Senior

Investigator award to Professor Williamson (award number NF-SI_0513-10025). The

funders had no role in study design, data collection and analysis, decision to publish, or preparation of the manuscript."

"EG and PRW are members of the COMET Management Group. SG and KMS have

declared that no competing interests exist."

Reviewers' comments:

Reviewer's Responses to Questions

**Comments to the Author**

1. Is the manuscript technically sound, and do the data support the conclusions?

Reviewer #1: No

Reviewer #2: Yes

2. Has the statistical analysis been performed appropriately and rigorously? 

Reviewer #1: No

Reviewer #2: No

3. Have the authors made all data underlying the findings in their manuscript fully available?

Reviewer #1: Yes

Reviewer #2: Yes

4. Is the manuscript presented in an intelligible fashion and written in standard English?

Reviewer #1: No

Reviewer #2: Yes

5. Review Comments to the Author

Reviewer #1: I enjoyed reading this but I have a problem with it. It isn't what it purports to be in my view. It is not - in any sense - a "systematic review" as that term is usually understood. It is in effect an Annual Report on newly published course outcome sets for research. There is then some limited comparison of those published in 2019 (the subject of this paper) with those published earlier. But where these comparisons are made, no attempt is made at any statistical analysis. What I mean by that is that the differences observed (between this year's data and pervious years) may have occurred by chance alone. But this possibility is not measured, nor is it even entertained. It is assumed that the differences are "real" and conclusions drawn or conjectures made.

There is a second, distinct piece of work embedded in this paper. This is a study looking at COS developed for conditions in children. This could be a stand alone piece of work

The third, equally distinct project is a brief look at what proportion of the most prevalent global diseases and injuries are 'covered' by a COS. The facts as reported will be interesting to some people. Perhaps. But this is really only a 'reporting' exercise.

Overall, I don't see this as a coherent piece of research. It's a Report. I could see it being published by COMET on their website.

What would it have to be to be more than that? Well, it would have to answer an important question. Such as - how has the development of COS improved over time? Has the quality of the methods used for such development improved? Has the quality of reporting of those methods improved? As it stands, the paper simply doesn't answer a clear question.

Reviewer #2: This is an updated review of articles that report the development of a core outcome set (COS). The authors report the number and quality of new COSs that have been developed in 2019. In addition, the authors report that 16% of 68 COSs that concern children in the Comet database included children in the development-process and still 6 of 25 globally prevalent diseases do not have a COS.

General comments: The COMET database is an important and interesting project and it is interesting to see how the COSs are developing. It is a bit difficult to see the difference between the update and the extra review of the COMET database. For the update it seems that the authors want to show the timeline of COSs development, but they don’t mention this in the abstract. It is also difficult to judge the timeline as the comparison what is already there (447 COSs?) comes only half way the results. It is unclear how the 2019 increase relates to the other increases and what the authors expected.

More detailed comments

- In the abstract mention the total amount of COSs in the Comet database and the timeline.

- In the methods, better separate the COSs update, the exploration of child involvement and COSs for prevalent diseases. Please write children and young adults in full or just define and use children. CYP is used for cytochrome P-450. In general, it is not clear what is meant with young persons. Is it young adults or does this include persons younger than 18?

- It is unclear how the minimum standards for COS development were judged. The authors say the assessed twelve criteria, but I assume that domains are meant. For each domain there should be criteria that discern compliance from non-compliance. What are for example the criteria for using unambiguous language?

- The use of only the 25% top ranked references coming up in the search sounds like am arbitrary cut-off point. Now that the authors have used this automated search several times, could they reflect on the efficiency and accuracy of it? When there are 25 out of 81 records that do not come up in the search and that are deemed important, the search does not seem to be very sensitive.

- It would be good to indicate uncertainty in the findings. For example, when the authors report that 39% of the studies report using COS reporting guideline and stating that this is less than the 60% previously reported.

- Six of the most prevalent diseases do not have a COS. Can the authors indicate mechanisms by which this situation could improve?

- The authors conclude that the production of COS remains consistently high. This depends on the number to be achieved. Do the authors have an idea how many COSs would be needed and can they relate the findings to this number?

6. PLOS authors have the option to publish the peer review history of their article (what does this mean?). If published, this will include your full peer review and any attached files.

Reviewer #1: No

Reviewer #2: **Yes: **Jos Verbeek

---

## [Author Response · Author response to Decision Letter 0]

18 Nov 2020

Journal Requirements:

Response: We have checked that the manuscript matches style requirements and made any necessary changes.

"Professor Williamson is a National Institute for Health Research (NIHR) Senior Investigator (award number NF-SI_0513-10025). The views expressed in this article are those of the author(s) and not necessarily those of the NIHR, or the Department of Health and Social care. Karen Matvienko-Sikar is supported by a Health Research Board Applying Research into Policy and Practice Fellowship (award number HRB-ARPP-A-011)."

"This work was funded through a National Institute for Health Research (NIHR) Senior

Investigator award to Professor Williamson (award number NF-SI_0513-10025). The

funders had no role in study design, data collection and analysis, decision to publish, or preparation of the manuscript."

Response: Funding information have been removed from the manuscript. Please could you update the Funding Statement as follows: 

“Professor Williamson is a National Institute for Health Research (NIHR) Senior Investigator (award number NF-SI_0513-10025). Karen Matvienko-Sikar is supported by a Health Research Board Applying Research into Policy and Practice Fellowship (award number HRB-ARPP-A-011). The

funders had no role in study design, data collection and analysis, decision to publish, or preparation of the manuscript."

"EG and PRW are members of the COMET Management Group. SG and KMS have

declared that no competing interests exist."

Response: We have included our updated Competing Interests statement in our cover letter. 

Reviewer #1: I enjoyed reading this but I have a problem with it. It isn't what it purports to be in my view. It is not - in any sense - a "systematic review" as that term is usually understood. It is in effect an Annual Report on newly published course outcome sets for research. There is then some limited comparison of those published in 2019 (the subject of this paper) with those published earlier. But where these comparisons are made, no attempt is made at any statistical analysis. What I mean by that is that the differences observed (between this year's data and pervious years) may have occurred by chance alone. But this possibility is not measured, nor is it even entertained. It is assumed that the differences are "real" and conclusions drawn or conjectures made.

Response: We thank the reviewer for their comments. The purpose of a systematic review is to identify and synthesise all available primary evidence that meets pre-specified eligibility criteria to answer a specific research question. Our study complies with this definition as we are presenting the annual update of a systematic review by aiming to identify all COS development studies published in 2019. We describe the methodological approaches used in these studies to enable us to highlight areas for improvement in COS development. The primary aim of the review has been expanded to make this clear (see lines 107-109). No statistical analysis was performed, rather a narrative approach was undertaken to synthesise data. We have identified all published COS so are not presenting a sample but rather the population of studies. As we state in the manuscript, the objective of the review is “to describe current knowledge and trends about COS development methods to help inform discussion around what COS developers are currently doing; in both what they are doing consistently, what is becoming standard methods of development, what COS developers are doing well, improvements in methods of COS development, and where there might be gaps in health areas or where methodological research is needed” (see lines 76-81).

There is a second, distinct piece of work embedded in this paper. This is a study looking at COS developed for conditions in children. This could be a stand alone piece of work. The third, equally distinct project is a brief look at what proportion of the most prevalent global diseases and injuries are 'covered' by a COS. The facts as reported will be interesting to some people. Perhaps. But this is really only a 'reporting' exercise.

Response: The secondary aim, which relates to the inclusion of children in COS development studies is a related piece of work, and this additional data extraction complements the review. In recent years, the importance of including children in the COS development process has been emphasised. Therefore, we felt it was important to look back at all published COS development studies to explore their participation. Going forward we will continue to extract this data and will look to see whether the inclusion of children in COS development increases over time. The third aim, which relates to identifying COS applicable to the most prevalent global diseases is also relevant, as it serves to identify gaps where COS development work is needed. By including this piece of work in our manuscript, we can highlight the priority areas for COS development. 

Overall, I don't see this as a coherent piece of research. It's a Report. I could see it being published by COMET on their website. What would it have to be to be more than that? Well, it would have to answer an important question. Such as - how has the development of COS improved over time? Has the quality of the methods used for such development improved? Has the quality of reporting of those methods improved? As it stands, the paper simply doesn't answer a clear question.

Response: Our manuscript describes the current knowledge and trends about COS development methods and identifies research gaps in relation to the methodology of COS development and important health areas where COS do not exist. We have described above how the three aims of this research are related and have added additional detail throughout to enhance clarity. The data in supplementary tables 4-7 show how COS development has improved over time. Text has been added to the results section to highlight improvements (see lines 243-247).

Line 53: Does “these” refer to the studies, the COS or both? 

Response: We are referring to the studies and have now clarified this in the manuscript. 

Line 54: Systematic reviews are usually reviews of studies. It isn’t clear if “a systematic review of COS….” is (a) a systematic review of the studies reporting the development of COS (in which case one might expect judgements to be made about the qualities of those studies, as is usual in any other type of systematic review), or (b) more simply a catalogue of the COSs themselves. If it is (a) a better phrase would be “a systematic review of studies reporting the development of COSs” 

Response: The text has been edited. 

Line 55 – 62: The content of these lines suggests that it is important to review how COSs have been developed 

Response: The reviewer’s interpretation is correct. It is important to review how COS have been developed, which is why we assessed each of the published COS studies against the COS-STAD development standards (see Table 1). 

Line 77: Did they assess “each of the COS…” or did they assess the development of the COS, or more specifically the studies reporting the development of the COS? If the latter, there is always the possibility that the quality of the reporting (as distinct from the quality of the development itself) may affect their conclusions. 

Response: The text has been edited. We have assessed the development of the COS, but we acknowledge in the discussion section, that COS developers may not be reporting all of the information necessary to assess whether the standards have been met (see lines 290-293).

Line 84: The first aim – to identify new COSs published or indexed in 2019 – is clear and unambiguous. 

Response: Thank you.

Line 85: The question: have CYP been included as participants in the development of COSs relevant to CYP? is not an unreasonable one, but does not flow easily from the first. Nor the question about the methods used to facilitate their participation. I think these should be clearly listed as secondary aims of the project. 

Response: The text has been edited. 

Line: 86 The third aim is also distinct but more related to the first than the one relating to CYP. It is also the first mention of an attempt by the authors to look for “ongoing” COS development exercises. 

Response: The third aim relates to both published and ongoing COS studies. A sentence has been added to the introduction to make readers aware that the COMET database also includes details of ongoing COS development work (see lines 75-76). 

Line 88: The sentence beginning “Included COS ….” describes a “Method” 

Response: This sentence has been deleted.

Line 92: In any systematic review – even an update – the reader should expect (as a minimum) to know how the included studies were “located, appraised and synthesised”. The inclusion and exclusion criteria should be very clear, and listed bulleted form if necessary

Response: Additional details relating to the inclusion and exclusion criteria have now been added (see lines 122-138). 

Line 100: The sentence beginning “Studies were eligible for inclusion if…..” is better than the previous sentence which appears redundant. But a clear list would be better in my view. I can find no exclusion criteria 

Response: Exclusion criteria has been added (see lines 134-138). 

Line 135: I am finding it a bit wearrisome continually reading the phrase “as in the previous review update”. My view is that this submitted article should stand alone and the reader should not have to consult the previous papers to know what was done. An “updated” systematic review should be cumulative and provide an up-to-date review of the totality of the literature to date. This is quite different from saying – “here are the data from studies that result from searches covering the last 12 months”. 

Response: Edits have been made throughout to ensure the reader does not have to consult the previous papers to understand what was done in the current review. 

Line 144: The decision to extract information about CYP is interesting in the light of my last comments. The Authors could have gone back and extracted these data from all the studies identified in previous iterations of the review. It looks to me as if they have not done that. They are introducing a “change to protocol” here (and at least are being explicit about that, so I cannot criticise that) but this really is a separate thread of work compared with the main essence of the project, which is updating their review. 

Response: We have extracted information about the participation of children in all COS development studies published to date, including those identified in the previous reviews as well as the current update. Text has been added to make this clear (see lines 188-190). 

Line 166: I find myself wanting to know which are the 37 new COSs? Looking at ST Table, I cannot work this out. The Table focuses (in the first column) on the reports. Why not have the new COS’s in column 1 and the reports in a later column? Also, why not have a second Table with the studies identified which relate to COSs in the existing database?

Response: The studies linked to COS included in previous reviews are signified in the S2 Table [^] with a footnote. The 33 new studies are also signified as either [*] having considered outcomes while addressing wider clinical trial design issues (n=2); or [**] having specifically considered outcome selection and measurement (n=31). Three of the 33 new COS studies include >1 COS and footnotes have been added to signify these studies [a, b, c]. 

Reviewer #2: This is an updated review of articles that report the development of a core outcome set (COS). The authors report the number and quality of new COSs that have been developed in 2019. In addition, the authors report that 16% of 68 COSs that concern children in the Comet database included children in the development-process and still 6 of 25 globally prevalent diseases do not have a COS.

General comments: The COMET database is an important and interesting project and it is interesting to see how the COSs are developing. It is a bit difficult to see the difference between the update and the extra review of the COMET database. For the update it seems that the authors want to show the timeline of COSs development, but they don’t mention this in the abstract. It is also difficult to judge the timeline as the comparison what is already there (447 COSs?) comes only half way the results. It is unclear how the 2019 increase relates to the other increases and what the authors expected.

Response: Text has been edited to distinguish between the primary and secondary aims of this work and additional details has been added throughout the manuscript to enhance clarity. The objective of the review is not to provide a direct comparison of the 2019 COS studies with those that have previously been published, as the numbers are too small to compare. Rather, our manuscript describes the current knowledge and trends about COS development methods and identifies areas where methodological research is needed. In the introduction, we do state the following: “COS development is still a relatively new area of research, and therefore it is important to describe current knowledge and trends about COS development methods to help inform discussion around what COS developers are currently doing; in both what they are doing consistently, what are becoming standard methods of development, what COS developers are doing well, improvements in methods of COS development, and where there might be gaps in health areas or where methodological research is needed.” The data in supplementary tables 4-7 show how COS development has improved over time and text has been added to the results section to highlight improvements over time (see lines 241-247).

More detailed comments

- In the abstract mention the total amount of COSs in the Comet database and the timeline.

Response: Text has been added to the abstract (see lines 43-45).

- In the methods, better separate the COSs update, the exploration of child involvement and COSs for prevalent diseases. Please write children and young adults in full or just define and use children. CYP is used for cytochrome P-450. In general, it is not clear what is meant with young persons. Is it young adults or does this include persons younger than 18?

Response: We have used the PLoS One Level Headings to better separate the systematic review update, the exploration of child participation in COS and the COS priority setting work relating to the most prevalent global disease. In addition, we have replaced ‘CYP’ with ‘children’ throughout, as we are only referring to persons younger than 18. 

- It is unclear how the minimum standards for COS development were judged. The authors say the assessed twelve criteria, but I assume that domains are meant. For each domain there should be criteria that discern compliance from non-compliance. What are for example the criteria for using unambiguous language?

Response: The COS were assessed against the 11 minimum standards (a total of 12 criteria) as: yes (meeting that standard), no (not meeting that standard) or unsure (it was unclear whether the criteria had been met). The COS-STAD recommendations publication (Kirkham et al., 2027) provides explanatory information for each of the standards, which are used to determine whether or not a COS has complied with the standard. Regarding standard 11 (‘Care was taken to avoid ambiguity of language used in the list of outcomes’); the recommendations state the following: 

“The language used to describe each potential outcome for the core set should be unambiguous (standard 11). When considering language, adequate consideration should be given to getting this right for those involved in the consensus process as well as for potential users, which may lead to the use of both plain language descriptions and medical terms, with these pilot tested for understanding.”

- The use of only the 25% top ranked references coming up in the search sounds like am arbitrary cut-off point. Now that the authors have used this automated search several times, could they reflect on the efficiency and accuracy of it? When there are 25 out of 81 records that do not come up in the search and that are deemed important, the search does not seem to be very sensitive.

Response: The cut-off for screening was set to the top 25% of abstracts in ranked priority order, as per the evaluation of the automated screening model (Norman C, Gargon E, Leeflang M, Neveol A, Williamson PR. Evaluation of an automatic article selection method for timelier updates of the COMET Core Outcome Set database. Database 2019;2019(baz109). pmid:31697361). In the fourth update, we reported on the efficiency and accuracy of the automated screening model:

“By using the automated screening methods to rank citations, we screened 75% less abstracts (1649/5878 abstracts), and consequently screened 50% less full texts than in the previous update (274 compared to 514 in the previous update). We estimate this to have saved approximately 110 hours, with the screening taking only 73 hours. All abstracts excluded by automated ranking were found to be correctly excluded (Fig 1).”

The 25 record that did not come out in the search would not have been expected to, as these records predominantly consist of linked supplementary papers, which were identified from reading the main COS publication, which did come out in the search. There were six new COS studies included within the 25 records. The reasons they were not identified in our search was because they had either not been indexed in the bibliographic databases within the appropriate search timeframe or had been incorrectly excluded from an earlier review update. Text has been added to Figure 1 and the results section to clarify this (see lines 204-208). 

- It would be good to indicate uncertainty in the findings. For example, when the authors report that 39% of the studies report using COS reporting guideline and stating that this is less than the 60% previously reported.

Response: We have identified all published COS so are not presenting a sample but rather the population of studies, and describing those studies rather than testing for differences.

- Six of the most prevalent diseases do not have a COS. Can the authors indicate mechanisms by which this situation could improve?

Response: The current gaps in existing COS should be considered by COS developers to identify priority areas for COS development. By highlighting these important gaps in an open access publication, we hope to draw attention to this issue, which will hopefully lead to improvements. We will disseminate our findings to research funders who endorse consideration and fund development of COS (see https://www.comet-initiative.org/COSEndorsement/TrialFunders) by forwarding the link to the open access article. 

- The authors conclude that the production of COS remains consistently high. This depends on the number to be achieved. Do the authors have an idea how many COSs would be needed and can they relate the findings to this number?

Response: We agree and have removed the text which states that publication of COS remains high. 

There is no target number of COS to be achieved, as it depends whether or not there is a need for a COS to be developed ion a specific disease area. By including the assessment of the relevance of COS to the 25 leading causes of global disease burden, we have been able to highlight priority areas for COS development.

---

## [Editor Report · Decision Letter 1]

18 Dec 2020

Choosing important health outcomes for comparative effectiveness research: 6th annual update to a systematic review of core outcome sets for research

PONE-D-20-21185R1

Dear Dr. Gorst,

We’re pleased to inform you that your manuscript has been judged scientifically suitable for publication and will be formally accepted for publication once it meets all outstanding technical requirements.

Kind regards,

Charles S. Wiysonge, MD, PhD

Academic Editor

PLOS ONE
---

## [Editor Report · Acceptance letter]

2 Jan 2021

PONE-D-20-21185R1 

Choosing important health outcomes for comparative effectiveness research: 6^th ^ annual update to a systematic review of core outcome sets for research 

Dear Dr. Gorst:

I'm pleased to inform you that your manuscript has been deemed suitable for publication in PLOS ONE. Congratulations! Your manuscript is now with our production department. 

Kind regards, 

on behalf of

Professor Charles S. Wiysonge 

Academic Editor

PLOS ONE